# Prognostic implications of ultra-short heart rate variability indices in hospitalized patients with infective endocarditis

**Shay Perek**[1,2,3], **Udi Nussinovitch**[4,5], **Neta Sagi**[6], **Yori Gidron**[7], **Ayelet Raz-Pasteur**[1,3]*

1 Department of Internal Medicine A, Rambam Health Care Campus, Haifa, Israel, 2 Department of Emergency Medicine, Rambam Health Care Campus, Haifa, Israel, 3 The Ruth and Bruce Rappaport Faculty of Medicine, The Technion–Israel Institute of Technology, Haifa, Israel, 4 Department of Cardiology, Wolfson Medical Center, Holon, Israel, 5 Faculty of Medicine, Tel Aviv University, Tel Aviv, Israel, 6 Department of Pediatrics A, Rambam Health Care Campus, Haifa, Israel, 7 Department of Nursing, Faculty of Social Welfare and Health Sciences, University of Haifa, Haifa, Israel

* a_raz@rambam.health.gov.il

**Data Availability Statement:** The dataset generated and analyzed during the current study are available in the figshare repository, https://doi.org/10.6084/m9.figshare.21679742.v1.

## Abstract

### Background

Infective endocarditis (IE) is a disease that poses a serious health risk. It is important to identify high-risk patients early in the course of their treatment. In the current study, we evaluated the prognostic value of ultra-short heart-rate variability (HRV), an index of vagal nerve activity, in IE.

### Methods

Retrospective analysis was performed on adult patients admitted to a tertiary hospital due to IE. A logistic regression (LR) was used to determine whether clinical, laboratory, and HRV parameters were predictive of specific clinical features (valve type, staphylococcal infection) or severe short-term complications (cardiac, metastatic infection, and death). The accuracy of the model was evaluated through the measurement of the area under the curve (AUC) of the receiver operating characteristic curve (ROC). An analysis of survival was conducted using Cox regression. A number of HRV indices were calculated, including the standard deviation of normal heart-beat intervals (SDNN) and the root mean square of successive differences (RMSSD).

### Results

75 patients, aged 60.3(±18.6) years old, were examined. When compared with published age- and gender-adjusted HRV norms, SDNN and RMSSD were found to be relatively low in our cohort (75%-76% lower than the median; 33%-41% lower than the 2nd percentile). 26 (34.6%) patients developed a metastatic infection, with RMSSD<7.03ms (adjusted odds ratio (aOR) 9.340, p = 0.002), incorporated in a multivariate LR model (AUC 0.833). Furthermore, 27(36.0%) patients were diagnosed with Staphylococcus IE, with SDNN<4.92ms

**Funding:** The authors received no specific funding for this work.

**Competing interests:** The authors have declared that no competing interests exist.

(aOR 5.235, p = 0.004), a major component of the multivariate LR model (AUC 0.741). Multivariate Cox regression survival model, included RMSSD (HR 1.008, p = 0.012).

## Conclusion

SDNN, and particularly RMSSD, derived from ultra-short ECG recordings, may provide prognostic information about patients presenting with IE.

## Introduction

Infective endocarditis (IE) is associated with high mortality rates and severe complications, including embolic events, valvular destruction and arrhythmias [1]. Therefore, it is crucial to stratify patients' risks at an early stage of their treatment. IE's prognosis depends on clinical factors, lab testing, imaging findings, or a combination of them [2]. Several hematological, chemical, and inflammatory markers have been suggested for the evaluation of prognosis in patients with IE, including white blood cell count levels, C-reactive protein (CRP) levels, procalcitonin levels, neutrophil-lymphocyte ratios [3–5], in addition to complete blood count indices and N-terminal pro-B-type natriuretic peptide levels [6–8]. More complex assessments of cytokines and cell-derived macrovesicles may also have potential value in diagnosis and prognostication of IE patients [9–11]. Several studies have demonstrated that echocardiographic findings are effective predictors of mortality and embolic events [12]. Additionally, echo-guided decision-making for early surgery in IE patients has been shown to be a cost-effective management strategy [13]. Cardiac computed tomography may be an adjunctive test to echocardiography in patients with IE who are undergoing surgery [14].

While the majority of these methods are unavailable at the time of patient arrival, an electrocardiogram (ECG) is a routinely performed, non-invasive test, which is usually performed during the course of an emergency department (ED) visit. Electrocardiographic changes are prevalent in IE and are indicative of invasive disease, thereby have the ability to predict high morbidity and mortality [15]. Conduction abnormalities can also suggest that the disease is extending into the peri-valvular region [16] or signal the presence of complicated aortic valve disease [17, 18].

However, neuro-cardiac regulatory processes could also play a role in IE. The dynamic modulation of heart rate (HR) is considered a surrogate of the interaction between the sympathetic and parasympathetic nervous systems [19], and is quantified by the fluctuation or variability in the time intervals between normal heartbeats (i.e.; heart rate variability [HRV]). HRV decreases under stress, either emotional or physical, increases in rest and is considered an important noninvasive marker. It could be used to assess the function of the autonomic nervous system (ANS) as well as to determine if a cardiac response to autonomic modulation of heart rate is appropriate [20]. An inverse relation between HRV and inflammatory surrogates such as CRP has been established with 24 hour HRV. Moreover, HRV and CRP were found to have a synergistic effect in the prediction of death and myocardial infarction in ambulatory subjects with no apparent heart disease [21]. An increased mortality rate was associated with a lower HRV indices, calculated from short-term recordings, in patients with sepsis [22]. Similar findings were demonstrated in inflammatory bowel disease patients [23]. Interactions between autonomic innervation and immune responses are complex and intriguing. The vagus reflexively inhibits inflammation by activating the hypothalamic-pituitary-adrenal axis resulting with cortisol secretion [24]. Additionally, the vagus inhibits inflammation by a vagal-

to-sympathetic innervation of the spleen, where unique T-cells signal macrophages to stop synthesizing pro-inflammatory cytokines [25]. In contrast to its anti-inflammatory role, vagus nerve stimulation could also increase cytotoxic T-cells, as well as natural killer cells [26].

Ultra short HRV analysis, focuses on computation of HRV indices from recordings lasting less than 5 minutes [27]. Several studies have demonstrated strong correlations between certain ultra-short HRV indices and parameters derived from longer HRV recordings, especially for time-domain indices [28–34]. Additionally, several ultra-short HRV parameters have been shown to be prognostic in patients admitted with various cardiac conditions. For instance, in patients with ST segment elevation myocardial infarction, reduced 10-second HRV has been found to be an independent predictor of all-cause mortality [35]. Ultra short HRV indices were also found to poses prognostic potential in COVID-19 [36], as well as in myocarditis patients [37]. Yet, there is little information available about the HRV response in IE patients. Low 24-hour HRV has been proposed as a method for IE complication marker [38]. Reports on ultra-short electrocardiographic markers for ANS dysfunction in IE are lacking. Our study thus aimed to evaluate the prognostic implications of ultra-short HRV indices derived from admission ECGs in patients with IE.

## Materials and methods

### Study design and population

This retrospective analysis is based on Rambam health care campus (RHCC; Haifa, Israel) patient medical records. This study was performed in line with the principles of the Declaration of Helsinki. Approval was granted by the Ethics Committee of Rambam Health Care Campus on 7 February 2017, number 0603-16-RMB. Since all the data was retrospectively collected, the IRB waived the need for individual informed consent. The following ICD-9-CM diagnosis codes were screened for: 421.0 (Acute and subacute bacterial endocarditis), 421.1 (Acute and subacute infective endocarditis in diseases classified elsewhere), 421.9 (Acute endocarditis, unspecified), 424.90 (Endocarditis, valve unspecified, unspecified cause), 424.91 (Endocarditis in diseases classified elsewhere), 424.99 (Other endocarditis, valve unspecified), 036.42 (Meningococcal endocarditis), 098.84 (Gonococcal endocarditis), 112.81 (Candidal endocarditis), 115.04 (Infection by Histoplasma capsulatum, endocarditis), 115.14) Infection by Histoplasma duboisii, endocarditis), and 115.94 (Histoplasmosis, unspecified, endocarditis).

All patients 18 years and older, who were diagnosed with IE based on the modified Duke criteria [39], between January 2010 and June 2015, were considered. Exclusion criteria included: missing ECG or HRV record from their ED visit, ECGs with irregular heartbeats (any non-sinus rhythm including atrial fibrillation or flutter, premature beats) or low resolution.

### Data collection

All ED visits and hospital discharge letters from the study period, were screened for a diagnosis of IE, utilizing the MDClone (Beer-Sheva, Israel) computer software. This study included only patients who had undergone a 10-second resting ECG in the ED. In addition, all medical records were reviewed to identify and exclude potentially ineligible patients. Patients' medical history (including IE risk factors), presenting symptoms (including clinical components of the modified Duke criteria), ED vital signs as well as laboratory results (i.e. complete blood count, chemistry panel and blood cultures), were collected. All echocardiography reports were studied, and disease specific findings, including valve dysfunction or rupture, peri-valvular abscess,

as well as cardiac dysfunction were documented. Furthermore, date of death and IE disease specific complications were noted.

### ECG analysis and computation of ultra-short HRV indices

Participating patients underwent a 10-second resting ECG (LAN Green model, Norav Medical, Yokneam, Israel), while lying motionless in a supine position for at least 30 seconds. The ECG electrodes were placed in anatomical positions according to standard procedure. An ECG viewing program was used to visualize the resting ECG files (Resting ECG version 5.62, Norav Medical), PR interval duration and QRS interval duration were automatically measured, and QT interval duration was calculated based on the Bazzett equation. Later, ECG files were analyzed with a custom version of the HRV analysis software able to import and analyze 10-second long recordings (HRV version 5.62, Norav Medical). These programs allowed for the automatic computation of HRV parameters. In addition, ECGs were manually checked and recordings with disturbances (e.g.; excessive noise, sudden baseline instability or low resolution), were excluded. ECGs which contained excessive premature ventricular or supraventricular activity (e.g.; atrial fibrillation, atrial flutter, atrial tachycardia) as well as advanced atrioventricular conduction abnormalities (e.g.; second degree atrioventricular blocks, complete heart block or other high degree AV conduction abnormalities) were also excluded. HRV linear time-domain variables (e.g.; standard deviation of RR intervals [SDNN] and root mean square successive differences [RMSSD]) were the focus of our study in light of their highest association with long-term recordings [33].

### Endpoints and evaluated clinical features

The collected clinical features included age, gender, predispositions such as the presence of a prostatic valve, systolic and diastolic blood pressure (BP), resting heart rate, O2 saturation, core body temperature, white blood cells (WBC) count, circulatory levels of Neutrophils, Lymphocytes, Platelets, Hemoglobin and serum creatinine levels. The resting short-term resting heart rate has been shown to provide a crude estimate of the HRV parameters over a 5-minute period, although the effectiveness varies depending on the parameter measured [31].

Study outcomes included IE complications—cardiac complications (e.g.; new heart failure with reduced ejection fraction [HFrEF; based on echocardiographic findings], arrhythmia, abscess); metastatic infection (e.g.; emboli, abscess and mycotic aneurysm); and the need for valvular surgical intervention. As staphylococcus infection is considered a more severe condition [40], the eventual isolation of this pathogen was also examined as a study endpoint. In addition, survival analysis during the follow-up period was carried out.

### Statistical analysis

The study database was analyzed with R software (version 4.0.3, The R Foundation for Statistical Computing, Vienna, Austria).

Descriptive statistics—continuous variables are presented as means with Standard Deviation (SD) or medians with interquartile range (IQR) and categorical variables are presented as percentages. Comparisons between groups were performed with Wilcoxon Rank Sum for continuous variables and Fisher's exact test for categorical variables.

Ultra-short HRV indices for each patient in the study cohort were compared with published normal value ranges, corrected for age and gender [41].

Correlations between variables and Boolean outcomes were examined with logistic regressions (LR) and presented as odds ratio (OR) with 95% confidence intervals (95%CI) and p-values. LR was carried out after verification of compliance with relevant assumptions (e.g.;

sufficiently large sample size, linearity of independent variables and log-odds, lack of strongly influential outliers and absence of multicollinearity). Variables found to have statistical significance (p-value<0.050) or trend (p-value<0.090) in univariate analyses, were introduced into a multivariate LR model, in a backward stepwise fashion. Multivariate model accuracy is presented with receiver operating characteristic (ROC) curves, including the area under the curve (AUC), with 95%CIs based on the bootstrapping method, as well as by the Hosmer and Lemeshow goodness-of-fit (HLGOF) and overall model p-value. HRV parameters were assessed as both continuous and Boolean variables (e.g. smaller than values corresponding to quartile 1 [Q1; 25th percentile]). Survival analysis was carried out with univariate and multivariate Cox regression and presented as hazard ratios (HR) with 95%Cis and p-values.

## Results

### Study population

In the initial evaluation, 229 patients were included based on ICD codes that corresponded to IE diagnoses. Upon additional review of patients' medical records, 45 patients were excluded as they did not fulfill the modified Duke criteria for IE diagnosis. An additional 75 patients did not have an ED ECG on the day of their admission to the hospital. 34 other patients had either a technically flawed ECG or an irregular rhythm. Thus, the final study sample included 75 patients, aged 60.3 (±18.6) years old. The majority (51, 68.0%) were men and 38 (50.7%) of the patients suffered from native valve IE.

Patient clinical, laboratory and electrocardiographic characteristics in patients with infected native and prosthetic valve, are presented in Table 1. Patients with prosthetic valve IE were significantly older, had lower heart rate, lower hemoglobin levels, longer PR interval and QTc. Notably, time domain HRV indices did not differ between native and prosthetic valve IE patients.

### HRV indices compared to published norms

HRV parameters were relatively low in the study cohort. Compared with published normal 10-second HRV values, corrected for age and gender [41], SDNN (median 7.72[ms], IQR 4.92 [ms] -21.42[ms]) was found to be lower than median values in 57 (76.0%) patients, with 31 (41.3%) patients having SDNN values lower than the 2nd percentile of their age and gender corrected range. As for RMSSD (median 11.8[ms], IQR 7.03[ms]– 28.18[ms]), 56 (74.6%) patients and 25 (33.3%) patients had values lower than age and gender corrected median and 2nd percentile, respectively.

### HRV indices and study outcomes

SDNN and RMSSD were found to be statistically significant lower in patients who developed HFrEF or metastatic infections. SDNN was also reduced (statistical trend) in patients ultimately diagnosed with Staphylococcus IE (Table 2). While analysis of HRV indices with regard to the development of a new arrhythmia yielded statistically insignificant results, stratification based on the type of arrhythmia (tachyarrhythmia vs. bradyarrhythmia) was also carried out. RMSSD in the tachyarrhythmia sub-group (median 65.0[ms], IQR 16.9[ms]-120.3[ms]) was high compared to RMSSD in the bradyarrhythmia sub-group (median 12.0[ms], IQR 5.6[ms]-16.7[ms]). Due to this heterogeneity, additional correlations were not conducted for this outcome.

**Table 1. Patient clinical, laboratory and electrocardiographic characteristics (native valve IE vs. prosthetic valve IE).**

| | Native valve IE (n = 38) | Prosthetic valve IE (n = 37) | p-value |
|---|---|---|---|
| Age (years) | 53.5 (37.0–64.5) | 73.0 (63.0–81.0) | <0.001* |
| Male gender (%) | 27 (71.0) | 24 (64.8) | 0.626 |
| **Emergency department vital signs** | | | |
| Systolic BP (mmHg) | 129.0 (117.2–138.7) | 126.0 (111.0–140.0) | 0.622 |
| Diastolic BP (mmHg) | 72.5 (62.0–81.7) | 67.0 (63.0–76.0) | 0.138 |
| Heart rate (bpm) | 100.0 (90.0–116.7) | 83.0 (77.0–96.0) | <0.001* |
| Saturation (%) | 96.0 (92.2–97.7) | 96.0 (94.0–97.0) | 0.864 |
| Body temperature (˚C) | 37.0 (36.7–37.4) | 36.9 (36.6–37.4) | 0.413 |
| **Emergency department labs** | | | |
| WBC (×$10^9$/L) | 11.0 (8.6–15.2) | 11.2 (9.3–14.3) | 0.987 |
| Hemoglobin (g/dL) | 11.9 (10.5–13.1) | 10.8 (10.0–11.7) | 0.036* |
| Platelets (×$10^9$/L) | 184.0 (122.5–265.0) | 175 (144.0–228.7) | 0.833 |
| Creatinine (mg/dL) | 1.1 (0.9–1.4) | 1.3 (0.9–1.5) | 0.337 |
| **ECG** | | | |
| PR interval (ms) | 156.0 (143.0–181.0) | 214.0 (168.0–250.0) | <0.001* |
| QRS duration (ms) | 90.0 (82.0–99.5) | 94.0 (82.0–106.0) | 0.487 |
| QTc interval (ms) | 421.0 (397.7–451.5) | 453.0 (424.0–496.0) | 0.004* |
| **HRV** | | | |
| SDNN (ms) | 7.7 (4.6–21.6) | 7.7 (5.4–19.6) | 0.685 |
| RMSSD (ms) | 9.8 (6.9–24.6) | 11.9 (7.4–33.7) | 0.517 |

median (IQR) or number (percentage of group)

BP–blood pressure; bpm–beats per minute; ECG–electrocardiography; SDNN—standard deviation of NN intervals; RMSSD—root mean square of successive RR interval differences; WBC–white blood cells; QTc–heart rate corrected QT interval

* P-value<0.05

## Heart failure with reduced ejection fraction

Six-teen patients (21.3%) were diagnosed with HFrEF, Univariate LR correlations are presented in Table 3. Patients who had developed HFrEF, were found to have higher heart rates on ED arrival as well as HRV in the lower quartile (SDNN<4.92ms, OR 4.36, p-value 0.014; RMSSD<7.03ms, OR 3.04, p-value 0.062). However, for predicting HFrEF, the multivariate LR model did not include any HRV parameter and was comprised of only initial heart rate on ED arrival (OR 1.05 [95%CI 1.01–1.08], p-value 0.002; ROC AUC 0.736).

## Metastatic infection

Twenty-six (34.6%) patients developed a metastatic infection during their index admission (23 metastatic emboli, 2 metastatic abscess and 1 –mycotic aneurysm). Patients who had been diagnosed with a metastatic infection, were found to be younger, with native valve IE, higher heart rate and body temperature on ED arrival and lower initial platelet levels. In addition, these patients had HRV in the lower quartile (SDNN<4.92ms, OR 3.75, p-value 0.016; RMSSD<7.03ms, OR 5.14, p-value 0.003), as described in Table 4. Multivariate LR model included 3 parameters: native valve IE (aOR 8.21 [95%CI 2.41–35.77], p-value 0.001), ED body temperature upon arrival (aOR 2.56 [95%CI 1.17–6.46], p-value 0.026) and RMSSD<7.03ms (aOR 9.34 [95% CI 2.46–44.67], p-value 0.002). Multivariate LR model ROC curve is presented in Fig 1. AUC was 0.83 (95%CI 0.74–0.92), with HLGOF p-value of 0.487 and overall model p-value <0.0001. Notably, RMSSD in the lower quartile had the strongest aOR in this multivariate classification model.

**Table 2.  HRV indices with relation to study outcomes.**

| | Outcome | | p-value |
|---|---|---|---|
| | HRrEF (n = 16) | No HRrEF (n = 59) | |
| SDNN (ms) | 5.0 (4.1–7.7) | 8.8 (6.0–28.2) | 0.017* |
| RMSSD (ms) | 7.6 (5.0–12.2) | 12.1 (7.6–35.3) | 0.023* |
| | Peri-valvular Abscess (n = 8) | No Peri-valvular Abscess (n = 67) | |
| SDNN (ms) | 5.9 (3.8–9.0) | 7.8 (5.2–23.3) | 0.205 |
| RMSSD (ms) | 8.1 (3.9–11.8) | 12.0 (7.4–32.0) | 0.130 |
| | Arrhythmia (n = 13) | No Arrhythmia (n = 62) | |
| SDNN (ms) | 7.7 (4.2–67.2) | 7.7 (5.0–15.8) | 0.623 |
| RMSSD (ms) | 12.5 (7.1–83.6) | 11.4 (7.0–20.2) | 0.450 |
| | Metastatic Infection (n = 26) | No Metastatic Infection (n = 49) | |
| SDNN (ms) | 5.7 (4.0–13.1) | 8.4 (6.4–23.4) | 0.025* |
| RMSSD (ms) | 7.6 (4.6–20.2) | 12.5 (8.2–33.7) | 0.018* |
| | Valvular Surgery (n = 21) | No Valvular Surgery (n = 54) | |
| SDNN (ms) | 6.9 (4.6–9.5) | 8.2 (5.5–28.3) | 0.304 |
| RMSSD (ms) | 8.3 (5.7–13.0) | 11.9 (7.4–36.1) | 0.293 |
| | Staphylococcus Infection (n = 27) | No Staphylococcus Infection (n = 48) | |
| SDNN (ms) | 5.4 (4.0–11.7) | 8.6 (6.3–23.2) | 0.065† |
| RMSSD (ms) | 7.8 (5.2–17.4) | 12.0 (8.0–31.2) | 0.147 |

median (IQR)

HFrEF–heart failure with reduced ejection fraction; SDNN—standard deviation of NN intervals; RMSSD—root mean square of successive RR interval differences

* P-value<0.05

† P-value<0.09

## Valve surgery

Twenty-one (28.0%) IE patients required valvular surgery. HRV indices were not found to be correlated with the need for valve surgery, with only patient age found to have a statistically significant correlation (OR 0.96 (95%CI 0.93–0.99); p-value 0.017).

## Staphylococcus IE

Twenty-seven (36.0%) patients were diagnosed with Staphylococcus IE. SDNN<4.92ms was found to have statistically significant univariate correlation with Staphylococcus IE (OR 4.68; p-value 0.006), as presented in Table 5. Multivariate LR model included 2 parameters: ED creatinine (aOR 2.47 [95%CI 1.08–6.29], p-value 0.039) and SDNN<4.92ms (aOR 5.23 [95%CI 1.70–17.49], p-value 0.004). Multivariate LR model ROC curve is presented in Fig 2. AUC was 0.741 (95%CI 0.622–0.860), with HLGOF p-value of 0.585 and overall model p-value 0.001. Noticeably, SDNN in the lower quartile had the strongest aOR in the multivariate classification model.

## Survival analysis

Forty-four patients (58.6%) died within the study follow-up period (median follow-up 5.06 years, IQR 0.20–8.84 years). SDNN (HR 1.01 [95%CI 1.00–1.02], p-value 0.010) and RMSSD (HR 1.00 [1.00–1.01], p-value 0.012) were both found to have univariate statistically significant Cox regression relations (Table 6).

Multivariate Cox regression survival model, included 3 parameters: ED hemoglobin (HR 0.79 [95%CI 0.66–0.95], p-value 0.014), ED creatinine (HR 2.13 [95%CI 1.38–3.28], p-value

**Table 3. Development of new HFrEF–univariate logistic regression.**

| | Odds ratio | 95% Confidence interval | | p-value |
|---|---|---|---|---|
| | | Low | High | |
| Age (years) | 0.97 | 0.94 | 1.00 | 0.085† |
| Male gender | 2.39 | 0.61 | 9.36 | 0.209 |
| Native valve | 1.33 | 0.43 | 4.04 | 0.615 |
| Systolic BP (mmHg) | 0.99 | 0.97 | 1.02 | 0.863 |
| Diastolic BP (mmHg) | 0.98 | 0.93 | 1.03 | 0.589 |
| Heart rate (bpm) | 1.05 | 1.01 | 1.08 | 0.002* |
| O2 saturation (%) | 0.99 | 0.95 | 1.03 | 0.762 |
| Body temperature (C˚) | 0.93 | 0.44 | 1.98 | 0.865 |
| WBC (×$10^9$/L) | 1.06 | 0.96 | 1.17 | 0.195 |
| Neutrophils (×$10^9$/L) | 1.06 | 0.96 | 1.18 | 0.207 |
| Lymphocytes (×$10^9$/L) | 1.25 | 0.53 | 2.94 | 0.608 |
| Hemoglobin (g/dL) | 0.96 | 0.72 | 1.28 | 0.802 |
| Platelets (×$10^9$/L) | 0.99 | 0.99 | 1.00 | 0.537 |
| Creatinine (mg/dL) | 1.88 | 0.80 | 4.38 | 0.141 |
| SDNN (ms) | 0.97 | 0.93 | 1.01 | 0.157 |
| RMSSD (ms) | 0.97 | 0.93 | 1.00 | 0.142 |
| SDNN<4.92ms | 4.36 | 1.34 | 14.18 | 0.014* |
| RMSSD<7.03ms | 3.04 | 0.94 | 9.85 | 0.062† |

BP–blood pressure; bpm–beats per minute; SDNN—standard deviation of NN intervals; RMSSD—root mean square of successive RR interval differences; WBC–white blood cells

* P-value<0.05

† P-value<0.09

0.0005), RMSSD (HR 1.00 [95%CI 1.00–1.01], p-value 0.012). An increase in HRV indices was found to be associated with reduced survival over time, in both univariate and multivariate analyses.

## Discussion

In this retrospective study, we have demonstrated, for the first time, that the use of ultra-short time domain HRV indices, collected upon arrival to the ED, can predict morbidity and mortality in IE patients. HRV indices were found to be lower in our cohort, when compared with published age and gender adjusted norms. 75%-76% of patients had HRV values lower than median norms and 33%-41% were found to have HRV levels lower than the normal 2nd percentile. The SDNN and the RMSSD are the primary time-domain measure used to estimate the vagally mediated changes reflected in HRV [42]. In our study, we have demonstrated independent associations between RMSSD, SDNN and IE complications as well as with survival. Of note, HRV indices did not differ significantly between patients with native versus prosthetic valve IE.

IE patients who had developed systemic emboli, were found to have lower RMSSD values upon their ED presentation. Low RMSSD has been established as a negative prognostic index in several patient populations. In a group of adult bone marrow transplant patients, the investigators observed a significant reduction in RMSSD prior to the clinical diagnosis and treatment of sepsis [43]. Similar findings regarding the prognostic role of RMSSD, have been demonstrated in oncologic [44, 45] and post myocardial infarction patients [46].

**Table 4. Metastatic infection–univariate logistic regression.**

| | Odds ratio | 95% Confidence interval | | p-value |
|---|---|---|---|---|
| | | Low | High | |
| Age (years) | 0.97 | 0.94 | 0.99 | 0.033* |
| Male gender | 1.44 | 0.50 | 4.10 | 0.493 |
| Native valve | 5.74 | 1.94 | 16.93 | 0.001* |
| Systolic BP (mmHg) | 1.01 | 0.99 | 1.03 | 0.118 |
| Diastolic BP (mmHg) | 1.02 | 0.98 | 1.07 | 0.188 |
| Heart rate (bpm) | 1.02 | 1.00 | 1.05 | 0.031* |
| O2 saturation (%) | 0.99 | 0.96 | 1.03 | 0.959 |
| Body temperature (C˚) | 2.10 | 1.06 | 4.17 | 0.032* |
| WBC ($\times 10^9$/L) | 1.01 | 0.93 | 1.10 | 0.738 |
| Neutrophils ($\times 10^9$/L) | 1.01 | 0.91 | 1.11 | 0.826 |
| Lymphocytes ($\times 10^9$/L) | 0.79 | 0.36 | 1.74 | 0.571 |
| Hemoglobin (g/dL) | 1.09 | 0.86 | 1.40 | 0.448 |
| Platelets ($\times 10^9$/L) | 0.99 | 0.98 | 0.99 | 0.049* |
| Creatinine (mg/dL) | 0.88 | 0.39 | 1.98 | 0.772 |
| SDNN (ms) | 0.98 | 0.96 | 1.00 | 0.177 |
| RMSSD (ms) | 0.99 | 0.97 | 1.00 | 0.210 |
| SDNN<4.92ms | 3.75 | 1.26 | 11.13 | 0.016* |
| RMSSD<7.03ms | 5.14 | 1.69 | 15.62 | 0.003* |

BP–blood pressure; bpm–beats per minute; SDNN—standard deviation of NN intervals; RMSSD—root mean square of successive RR interval differences; WBC–white blood cells

* P-value<0.05

On the other hand, our study findings have demonstrated that an increase in RMSSD, as a continuous variable, is correlated with mortality over time. However, the magnitude of this association was weak. Several studies have shown correlations between high RMSSD and patient negative outcome, in other infectious diseases. An increased RMSDD, was documented in patients who developed septic shock within a few hours of presentation at the ED [47]. RMSDD was also found to be significantly increased among COVID-19 patients compared to healthy controls matched for age, gender and comorbidities [48]. Finally, in patients suffering from bacterial pneumonia, increased RMSSD obtained form 10-second ECGs has been associated with decreased survival [49].

In an analysis of HRV in patients undergoing peritoneal dialysis, higher RMSSD was associated with increased mortality [50]. Furthermore, in a large cohort of patients with chronic kidney disease, both very low and very high RMSDD were associated with increased risk for all-cause mortality [51]. These findings are in line with our observation that high RMSSD may predict a higher risk of death in patients with IE. The common factor in all these studies, may be that patients with high vagal activity at arrival to the hospital, could already have an infectious or inflammatory condition, to which the cholinergic anti-inflammatory vagus may respond. It is possible that, in the context of an infectious disease, sub-clinically elevated inflammation could cause an increase in vagal activity, indexed by high HRV, to activate the cholinergic anti-inflammatory reflex, and such a compensatory increase in HRV in such contexts, may then predict poor prognosis in infectious diseases such as IE. However, given our relatively small sample size and the small effect size of this HRV-survival relation, this requires replication in larger studies, together with testing this proposed compensation mechanism.

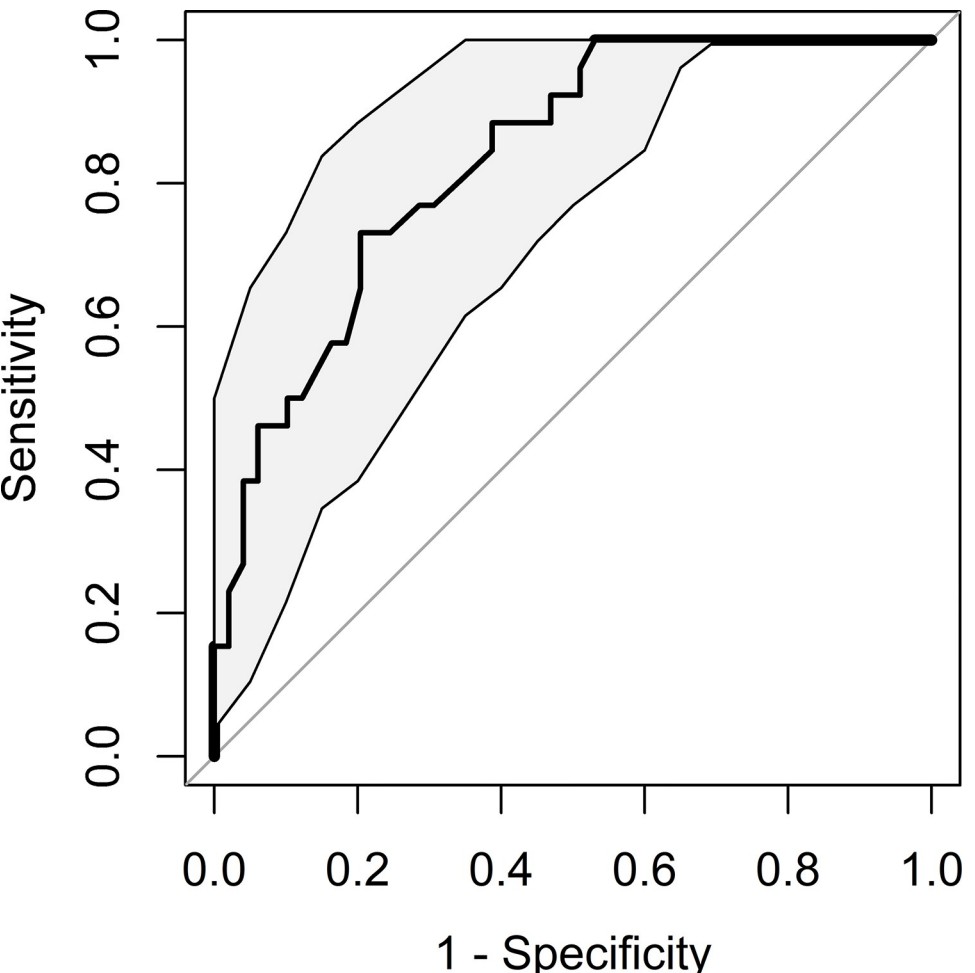

**Fig 1. Infective endocarditis metastatic infection–multivariate logistic regression prediction model receiver operating characteristic curve, with 95% confidence interval.**

Low SDNN values were correlated with Staphylococcal IE, in both univariate and multivariate LR classification. Staphylococcal infection has been identified as a risk factor for embolic events in IE [52]. In our study, we have demonstrated the association between low HRV indices and both embolic events and Staphylococcal IE, indicating these known reciprocal relations may be identified by HRV parameters.

Similar to our findings, low SDNN has also been correlated with infectious disease complications. In children diagnosed with viral myocarditis, lower SDNN predicted the development of ventricular arrhythmia [53].

Echocardiography has a significant role in the risk stratification of endocarditis patients, by providing information correlated with negative outcomes, such as a large vegetation, para-valvular infection, signs of increased left-cavities filling pressures, pulmonary hypertension and low left ventricular ejection fraction [54]. At admission, early assessment of prognosis is critical and has the ability to identify high risk patients, who may need closer monitoring, require more aggressive treatment (e.g. early surgery) and who may benefit from more frequent follow-up assessments. Serial echocardiographic examinations allow close follow up of endocarditis patients during antibiotic therapy and after disease resolution. Echocardiogram

**Table 5. Staphylococcus IE–univariate logistic regression.**

| | Odds ratio | 95% Confidence interval | | p-value |
|---|---|---|---|---|
| | | Low | High | |
| Age (years) | 0.99 | 0.96 | 1.01 | 0.473 |
| Male gender | 2.10 | 0.71 | 6.17 | 0.177 |
| Native valve | 1.35 | 0.52 | 3.50 | 0.525 |
| Systolic BP (mmHg) | 0.98 | 0.96 | 1.00 | 0.203 |
| Diastolic BP (mmHg) | 0.97 | 0.93 | 1.01 | 0.283 |
| Heart rate (bpm) | 1.01 | 0.99 | 1.04 | 0.164 |
| O2 saturation (%) | 0.98 | 0.94 | 1.02 | 0.349 |
| Body temperature (C°) | 1.53 | 0.81 | 2.87 | 0.183 |
| WBC ($\times10^9$/L) | 1.08 | 0.99 | 1.18 | 0.080† |
| Neutrophils ($\times10^9$/L) | 1.09 | 0.98 | 1.20 | 0.081† |
| Lymphocytes ($\times10^9$/L) | 0.53 | 0.22 | 1.22 | 0.140 |
| Hemoglobin (g/dL) | 0.89 | 0.69 | 1.15 | 0.391 |
| Platelets ($\times10^9$/L) | 0.99 | 0.99 | 1.00 | 0.694 |
| Creatinine (mg/dL) | 2.24 | 0.99 | 5.09 | 0.051† |
| SDNN (ms) | 1.00 | 0.98 | 1.01 | 0.943 |
| RMSSD (ms) | 1.00 | 0.99 | 1.01 | 0.754 |
| SDNN<4.92ms | 4.68 | 1.55 | 14.13 | 0.006* |
| RMSSD<7.03ms | 2.54 | 0.87 | 7.39 | 0.085† |

BP–blood pressure; bpm–beats per minute; SDNN—standard deviation of NN intervals; RMSSD—root mean square of successive RR interval differences; WBC–white blood cells

* P-value<0.05

† P-value<0.09

frequency and type depend on the clinical presentation, the involved pathogen and initial echocardiographic findings on admission [55].

In addition to echocardiography, PET-CT is another tool which has recently been established for the diagnosis and follow up of IE, and can assist the early detection of its complications such as systemic embolism [56]. Following our research, we suggest considering more frequent echocardiography studies, as well as early PET-CT for endocarditis patients, with HRV indices associated with high risk of uncontrolled infection, systemic embolism, heart failure and increased mortality, regardless of their clinical condition.

The results of our study indicate a simple and readily available prognostic factor, namely HRV, which can be simply derived from brief 10-seconds ECGs or even from fingertip photoplethysmography sensors. Nonetheless, this study has limitations, which are related to its retrospective cohort design and also to its possible selection bias. Several patients were excluded from our analysis due to incorrect IE diagnosis and missing or inadequate ECGs. By only including patients with IE diagnosis based on the fulfilment of the Duke criteria with a high-quality ECG tracing of regular cardiac rhythm, we created a smaller yet more accurate IE patient cohort with valid ultra-short HRV indices. By focusing on patients enrolled in a tertiary referral medical center for 5.5 years and applying various statistical tests, which included adjustment for confounders, we have attempted to limit the consequences of a potential selection bias. In spite of this, we cannot predict if a larger study and a longer follow-up period would provide different results, and as such, future research should be conducted to address this question.

In addition, we included ultra-short ECGs for measuring HRV. These could not enable us to derive reliably the frequency-domain parameters of HRV, such as high frequency and low

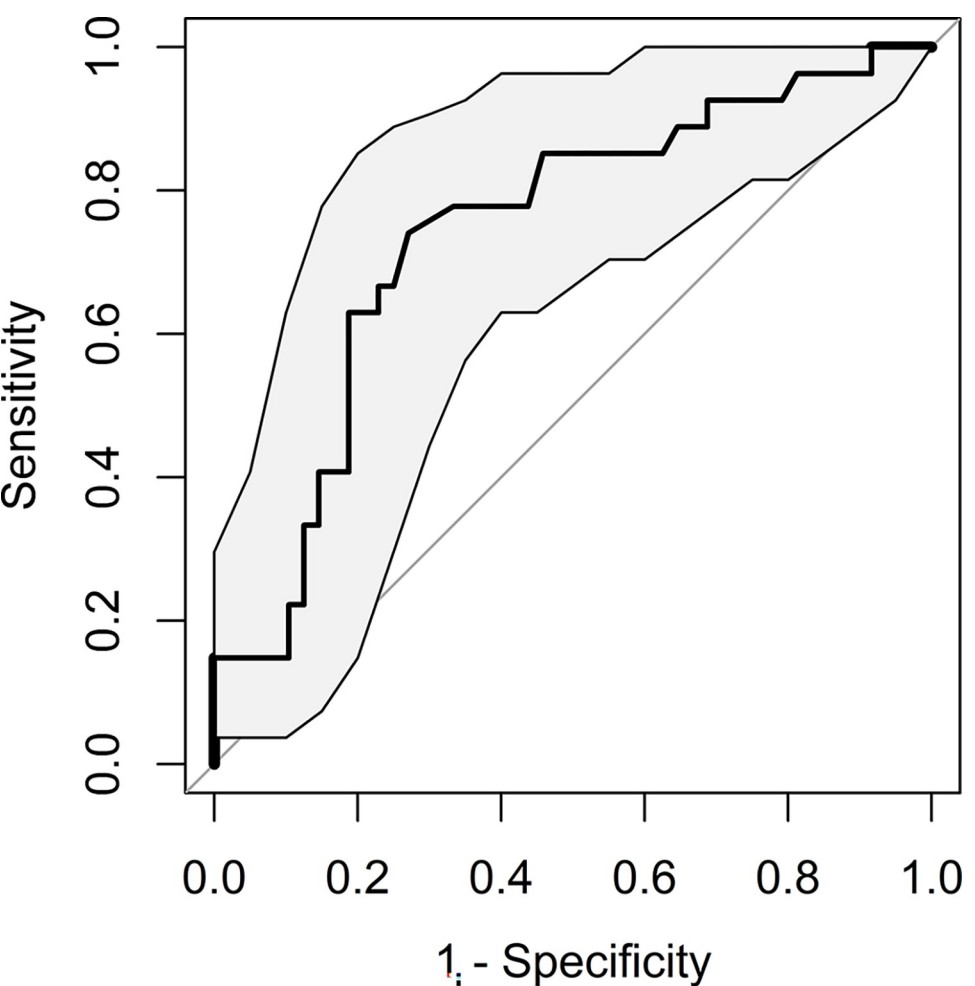

**Fig 2. Infective endocarditis Staphylococcus infection–multivariate logistic regression prediction model receiver operating characteristic curve, with 95% confidence interval.**

frequency HRV, which could potentially have additional prognostic value in IE. Furthermore, the exact length of time the patients laid down in a supine position prior to undergoing ECG is unknown. Nonetheless, even short stabilization periods prior to HRV assessment may be acceptable, especially in younger, healthier patients [57] and in static conditions [58] (similar to the setting in our study). Finally, due to the relatively small sample, a stratification to native and prosthetic valve IE did not yield significant results. Future prospective studies should address these issues, in order to validate our findings. Following our research, we suggest to consider more frequent radiographic studies (e.g.; PET-CT, echocardiogram) for endocarditis patients, with HRV indices associated with high risk of uncontrolled infection, systemic embolism and increased mortality, regardless of their clinical condition.

## Conclusions

Time domain ultra-short HRV indices based on ECG, carried out upon ED arrival, have the potential to allow for early risk stratification, in IE patients. Our findings should be validated, prior to implementation in patient management strategies.

**Table 6. Survival analysis–univariate cox regression.**

| | Hazard ratio (95%CI) | p-value |
|---|---|---|
| Age (years) | 1.02 (1.00–1.04) | 0.014* |
| Male gender | 0.70 (0.37–1.32) | 0.281 |
| Native valve | 0.78 (0.43–1.43) | 0.435 |
| Systolic BP (mmHg) | 1.01 (0.99–1.02) | 0.066† |
| Diastolic BP (mmHg) | 0.99 (0.96–1.02) | 0.758 |
| Heart rate (bpm) | 1.00 (0.98–1.01) | 0.646 |
| Saturation (%) | 0.98 (0.97–1.00) | 0.071† |
| Body temperature (˚C) | 1.02 (0.65–1.58) | 0.929 |
| WBC (×$10^9$/L) | 1.01 (0.96–1.06) | 0.560 |
| Neutrophils (×$10^9$/L) | 1.01 (0.96–1.07) | 0.539 |
| Lymphocytes (×$10^9$/L) | 0.37 (0.19–0.69) | 0.001* |
| Hemoglobin (g/dL) | 0.80 (0.67–0.96) | 0.018* |
| Platelets (×$10^9$/L) | 0.99 (0.99–0.99) | 0.030* |
| Creatinine (mg/dL) | 2.18 (1.46–3.25) | 0.0001* |
| SDNN (ms) | 1.01 (1.00–1.02) | 0.010* |
| RMSSD (ms) | 1.00 (1.00–1.01) | 0.012* |

BP–blood pressure; bpm–beats per minute; SDNN—standard deviation of NN intervals; RMSSD—root mean square of successive RR interval differences; WBC–white blood cells

\* P-value<0.05

† P-value<0.09

## Author Contributions

**Conceptualization:** Shay Perek, Udi Nussinovitch, Neta Sagi, Yori Gidron, Ayelet Raz-Pasteur.

**Data curation:** Shay Perek, Neta Sagi, Ayelet Raz-Pasteur.

**Formal analysis:** Shay Perek, Ayelet Raz-Pasteur.

**Investigation:** Shay Perek, Neta Sagi.

**Methodology:** Shay Perek, Udi Nussinovitch, Yori Gidron.

**Resources:** Udi Nussinovitch, Yori Gidron, Ayelet Raz-Pasteur.

**Software:** Shay Perek.

**Supervision:** Ayelet Raz-Pasteur.

**Validation:** Shay Perek, Ayelet Raz-Pasteur.

**Visualization:** Shay Perek.

**Writing – original draft:** Shay Perek, Udi Nussinovitch, Neta Sagi, Yori Gidron, Ayelet Raz-Pasteur.

**Writing – review & editing:** Shay Perek, Udi Nussinovitch, Neta Sagi, Yori Gidron, Ayelet Raz-Pasteur.

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
