## [Decision Letter · Decision Letter 0]

19 Apr 2023

PONE-D-23-02781Prognostic implications of ultra-short heart rate variability indices in hospitalized patients with infective endocarditisPLOS ONE

Dear Dr. Raz-Pasteur,

Thank you for submitting your manuscript to PLOS ONE. After careful consideration, we feel that it has merit but does not fully meet PLOS ONE’s publication criteria as it currently stands. Therefore, we invite you to submit a revised version of the manuscript that addresses the points raised during the review process.

As stated by both reviewers, please make sure that the statistical method is suitable for the nature of the data at hand and also please revise the result and discussion sections accordingly.

Also, please highlight the implications of your findings.

We look forward to receiving your revised manuscript.

Kind regards,

Atnafu Mekonnen Tekleab, M.D

Academic Editor

PLOS ONE

Journal Requirements:

https://www.mdpi.com/2077-0383/12/1/89/html

In your revision ensure you cite all your sources (including your own works), and quote or rephrase any duplicated text outside the methods section. Further consideration is dependent on these concerns being addressed.

Reviewers' comments:

Reviewer's Responses to Questions

**Comments to the Author**

1. Is the manuscript technically sound, and do the data support the conclusions?

Reviewer #1: Yes

Reviewer #2: Partly

2. Has the statistical analysis been performed appropriately and rigorously? 

Reviewer #1: Yes

Reviewer #2: No

3. Have the authors made all data underlying the findings in their manuscript fully available?

Reviewer #1: Yes

Reviewer #2: Yes

4. Is the manuscript presented in an intelligible fashion and written in standard English?

Reviewer #1: Yes

Reviewer #2: Yes

5. Review Comments to the Author

Reviewer #1: Dear Authors,

I compliment on your innovative work. Namely, as yourself declare, this is the first paper that addresses the topic of HRV as a prognosticator in patients with infective endocarditis.

A MEDLINE search yielded no results specifically on this topic, although there are published studies of HRV in other infectious and inflammatory conditions.

The other point of discussion is that you are using an ultra short HRV of 10 seconds, which is also a less frequently analyzed method when talking about HRV, where the most commonly published results for time domain analyzes are 5 minute SDNN analyses.

Given the structure of the written manuscript, after the clear introduction and nicely described pathophysiological mechanism of the role of the vagus in HRV mechanisms, as well as the clearly described methodology, the results section is a bit difficult to follow. There is no decisive statement for which of the monitored outcomes multivariate analysis is performed and for which not (due to the insufficient number of detected univariate predictors). It is necessary to read the text two or three times to understand it. The etiology (Staphylococcus) is not a subsequent outcome, but a feature that you have taken as a predictor that should be clearly emphasized.

Regarding the statistical power of the analyzed data, the insufficiency of the identified univariate predictors is a consequence of the small number of subjects (for such a strong analysis), and the small number of outcomes, especially perivalvular abscess-8 patients.

It is also debatable to create multivariate models consisting of only two variables (staphylococcal infection), or 3 variables (metastatic infection).

This observation of mine is best seen in the fatal outcome section where you have a powerful analysis, and where the debatable points about the statistical power of the analysis that I mentioned in the previous section are overcome. This is the strongest part of your manuscript.

Suggestion!

Please consider regrouping the outcomes (perivalvular abscess and metastatic infection basically have the same underlying mechanism - disseminated inflammation/infection!!!)

Finally, go through the text again for small typos, and pay attention to the brackets, they are not clearly placed. Use two types of brackets.

Reviewer #2: The research does try to add a new and easy way of assessing prognosis in IE patients in the way of HRV indices. However, I doubt its clinical application in the near future. Sample size is small as the authors state, and this will definitely affects the observed findings. I recommend major revision or explanation in the following areas:

1. More than half of IE patients were excluded. There is heavy selection bias, and I am concerned on the reliability & applicability of the findings presented.

2. The data analysis needs to be looked at again. Unlike what was described in the methods of data analysis, HRV parameters are instead presented using median values (not means), suggesting a skewed data. Sample size is small and data appears skewed, hence parametric tests such as Logistic regression may not be appropriate.

3. There appears some contradictory findings that are not fully explained. On one hand, lower HRV indices show predictive value in metastatic infections and Staph infection, and on the other increase in HRV predicts arrhythmic complications and mortality. Please explain.

6. PLOS authors have the option to publish the peer review history of their article (what does this mean?). If published, this will include your full peer review and any attached files.

Reviewer #1: **Yes: **Marija Vavlukis

Reviewer #2: No

---

## [Author Response · Author response to Decision Letter 0]

3 Jun 2023

Prof. Emily Chenette

Editor-in-chief, PLOS One

Re: PONE-D-23-02781

Thank you for your email of April 14, 2023, informing us that you are willing to reconsider acceptance of our paper for publication subject to revisions recommended by the reviewers.

Enclosed please find our revised manuscript entitled “Prognostic implications of ultra-short heart rate variability indices in hospitalized patients with infective endocarditis” by Shay Perek et al., marked with the changes made (as well as a clean version of the manuscript), and our detailed reply to the reviewers’ comments. 

We hope the paper will now be considered suitable for publication. Our point-by-point response to the reviewer's remarks is outlined below. 

Sincerely yours,

A. Raz-Pasteur, MD

Clinical Assistant Professor

Director, Internal Medicine 'A' Department

Rambam Health Care Campus

POB 9602 Haifa, Israel

Tel: 972-4-7773106

Fax: 972-4-7772721

E-mail: a_raz@rambam.health.gov.il

 

Editor:

Remark: As stated by both reviewers, please make sure that the statistical method is suitable for the nature of the data at hand and also please revise the result and discussion sections accordingly.

Response: Clarifications and corrections have been made to the methods, results and discussion sections.

Remark: Also, please highlight the implications of your findings.

Response: Following the mentioned corrections, our study provides insight into the use of ultra-short HRV indices as prognostic markers in infective endocarditis.

Remark: Please ensure that your manuscript meets PLOS ONE's style requirements

Response: The manuscript has been reviewed and the layout has been verified to be in accordance with the style requirements. 

Remark: We noticed you have some minor occurrence of overlapping text with the following previous publication(s), which needs to be addressed: https://www.mdpi.com/2077-0383/12/1/89/html

Response: Considerable revisions have been made to the few paragraphs that shared similarities with our previous paper, published in the Journal of Clin. Med. 

Remark: Your ethics statement should only appear in the Methods section of your manuscript. If your ethics statement is written in any section besides the Methods, please move it to the Methods section and delete it from any other section.

Response: Ethics statements have been removed from the 'Statements and Declarations' section and are now included in the Method section only. 

 

Reviewer 1

Remark: …after the clear introduction and nicely described pathophysiological mechanism of the role of the vagus in HRV mechanisms, as well as the clearly described methodology, the results section is a bit difficult to follow. There is no decisive statement for which of the monitored outcomes multivariate analysis is performed and for which not (due to the insufficient number of detected univariate predictors). It is necessary to read the text two or three times to understand it.

Response: The results section has been changed with an emphasis on logistic regression analysis for three outcomes (e.g.; heart failure with reduced ejection fraction, metastatic infection, Staphylococcal infection) as well as survival analysis.

Remark: The etiology (Staphylococcus) is not a subsequent outcome, but a feature that you have taken as a predictor that should be clearly emphasized.

Response: A change has been made to the results section to clarify that the likelihood of developing Staphylococcal infection was estimated not as an outcome, but rather as a clinical feature generally associated with a more severe illness. 

Remark: Regarding the statistical power of the analyzed data, the insufficiency of the identified univariate predictors is a consequence of the small number of subjects (for such a strong analysis), and the small number of outcomes, especially perivalvular abscess-8 patients.

It is also debatable to create multivariate models consisting of only two variables (staphylococcal infection), or 3 variables (metastatic infection). This observation of mine is best seen in the fatal outcome section where you have a powerful analysis, and where the debatable points about the statistical power of the analysis that I mentioned in the previous section are overcome. This is the strongest part of your manuscript. 

Please consider regrouping the outcomes (perivalvular abscess and metastatic infection basically have the same underlying mechanism - disseminated inflammation/infection!!!)

Response: The corrected results section includes logistic regression analysis only for outcomes with sufficient amount of observations. In case of a small number of subjects (e.g.; perivalvular abscess), only descriptive statistics is provided. 

Remark: Finally, go through the text again for small typos, and pay attention to the brackets, they are not clearly placed. Use two types of brackets.

Response: We appreciate the reviewer's comments. Numerous sentences were revised, and the manuscript was screened for typos.

 

Reviewer 2:

Remark: Sample size is small as the authors state, and this will definitely affect the observed findings. More than half of IE patients were excluded. There is heavy selection bias, and I am concerned on the reliability & applicability of the findings presented.

Response: In response to the reviewer's concern and to discuss the possible implications, we have expanded the limitations section of this study. It is true that ECGs are not routinely performed in emergency departments when patients are suspected of having infectious diseases such as IE. This study may be important because it highlights the valuable prognostic information that can be obtained from ECGs of patients with IE. ECGs should therefore be performed more frequently in patients with a predisposition to IE, any sign that suggests a seemingly new cardiac murmur, or any number of symptoms that might suggest IE. 

Remark: The data analysis needs to be looked at again. Unlike what was described in the methods of data analysis, HRV parameters are instead presented using median values (not means), suggesting a skewed data. Sample size is small and data appears skewed, hence parametric tests such as Logistic regression may not be appropriate.

Response: The results section has been corrected and includes both non-parametric group comparisons (of HRV indices with relation to study outcomes), as well as logistic regression, only if the regression assumptions have been met (e.g.; linearity assumption). The methods section has been amended to include the description of these assumptions. 

Remark: There appears some contradictory findings that are not fully explained. On one hand, lower HRV indices show predictive value in metastatic infections and Staph infection, and on the other increase in HRV predicts arrhythmic complications and mortality. Please explain.

Response: The results section has been revised with an emphasis on the arrhythmic complication. This outcome included both tachyarrhythmia and bradyarrhtyhmias. A sub-analysis of these two groups revealed major differences between HRV indices (as mentioned in the results section). Therefore, we decided not to carry out the logistic regression analysis for the arrhythmia outcome and provide only descriptive statistics and non-parametric comparisons. As to survival analysis, the discussion section has been expanded with proposed mechanistic explanations.

---

## [Editor Report · Decision Letter 1]

8 Jun 2023

Prognostic implications of ultra-short heart rate variability indices in hospitalized patients with infective endocarditis

PONE-D-23-02781R1

Dear Dr. Ayelet Raz-Pasteur,

We’re pleased to inform you that your manuscript has been judged scientifically suitable for publication and will be formally accepted for publication once it meets all outstanding technical requirements.

Kind regards,

Atnafu Mekonnen Tekleab, M.D

Academic Editor

PLOS ONE
---

## [Editor Report · Acceptance letter]

14 Jun 2023

PONE-D-23-02781R1 

Prognostic implications of ultra-short heart rate variability indices in hospitalized patients with infective endocarditis 

Dear Dr. Raz-Pasteur:

I'm pleased to inform you that your manuscript has been deemed suitable for publication in PLOS ONE. Congratulations! Your manuscript is now with our production department. 

Kind regards, 

on behalf of

Dr. Atnafu Mekonnen Tekleab 

Academic Editor

PLOS ONE